# Modified Glasgow Prognostic Score as a Predictor of Recurrence in Patients with High Grade Non-Muscle Invasive Bladder Cancer Undergoing Intravesical Bacillus Calmette–Guerin Immunotherapy

**DOI:** 10.3390/diagnostics12030586

**Published:** 2022-02-25

**Authors:** Matteo Ferro, Octavian Sabin Tătaru, Gennaro Musi, Giuseppe Lucarelli, Abdal Rahman Abu Farhan, Francesco Cantiello, Rocco Damiano, Rodolfo Hurle, Roberto Contieri, Gian Maria Busetto, Giuseppe Carrieri, Luigi Cormio, Francesco Del Giudice, Alessandro Sciarra, Sisto Perdonà, Marco Borghesi, Carlo Terrone, Evelina La Civita, Pierluigi Bove, Riccardo Autorino, Matteo Muto, Nicolae Crisan, Michele Marchioni, Luigi Schips, Francesco Soria, Daniela Terracciano, Rocco Papalia, Felice Crocetto, Biagio Barone, Giorgio Ivan Russo, Stefano Luzzago, Giuseppe Mario Ludovico, Mihai Dorin Vartolomei, Francesco Alessandro Mistretta, Vincenzo Mirone, Ottavio de Cobelli

**Affiliations:** 1Division of Urology, European Institute of Oncology, Milan IRCCS, 20141 Milan, Italy; gennaro.musi@ieo.it (G.M.); stefano.luzzago@ieo.it (S.L.); francescoalessandro.mistretta@ieo.it (F.A.M.); ottavio.decobelli@ieo.it (O.d.C.); 2I.O.S.U.D., George Emil Palade University of Medicine and Pharmacy, Science and Technology, 540142 Targu Mures, Romania; sabin.tataru@gmail.com (O.S.T.); mdvartolomei@yahoo.com (M.D.V.); 3Department of Oncology and Hematology-Oncology, Università degli Studi di Milano, 20133 Milan, Italy; 4Urology, Andrology and Kidney Transplantation Unit, Department of Emergency and Organ Transplantation, University of Bari, 70126 Bari, Italy; giuseppe.lucarelli@inwind.it; 5Department of Urology, Magna Graecia University of Catanzaro, 88100 Catanzaro, Italy; abufarhan.abdal@gmail.com (A.R.A.F.); cantiello@unicz.it (F.C.); damiano@unicz.it (R.D.); 6Department of Urology, Istituto Clinico Humanitas Istituto di Ricovero e Cura a Carattere Scientifico-Clinical and Research Hospital, Milan IRCCS, 20089 Milan, Italy; rodolfo.hurle@humanitas.it (R.H.); contieri.ro@gmail.com (R.C.); 7Department of Urology and Organ Transplantation, University of Foggia, 71122 Foggia, Italy; gianmaria.busetto@unifg.it (G.M.B.); giuseppe.carrieri@unifg.it (G.C.); 8Urology and Renal Transplantation Unit, Department of Medical and Surgical Sciences, University of Foggia, 71122 Foggia, Italy; luigi.cormio@unifg.it; 9Urology Unit, University of Foggia, Bonomo Teaching Hospital, 76123 Foggia, Italy; 10Department of Urology, Policlinico Umberto I, Sapienza University of Rome, 00185 Rome, Italy; francesco.delgiudice@uniroma1.it (F.D.G.); alessandro.sciarra@uniroma1.it (A.S.); 11Fondazione “G. Pascale” IRCCS, 80131 Naples, Italy; s.perdona@istitutotumori.na.it; 12Department of Urology, San Martino Policlinico Hospital, IRCCS for Oncology, 16132 Genoa, Italy; marco.borghesi@med.unipo.it (M.B.); carlo.terrone@med.unipo.it (C.T.); 13Department of Surgical and Integrated Diagnostic Sciences, University of Genoa, 16132 Genoa, Italy; 14Department of Translational Medical Sciences, University of Naples “Federico II”, 80131 Naples, Italy; eva.lacivita@gmail.com (E.L.C.); daniela.terracciano@unina.it (D.T.); 15Urology Department, San Carlo di Nancy Hospital, GVM Care and Research, 00165 Rome, Italy; pierluigi.bove@uniroma2.it; 16Urology Unit, Department of Surgery, Tor Vergata University of Rome, 00133 Rome, Italy; 17Division of Urology, VCU Health, Richmond, VA 23298, USA; ricautor@gmail.com; 18Radiotherapy Unit, “S.G. Moscati” Hospital, 74010 Avellino, Italy; mattomuto@gmail.com; 19Department of Urology, University of Medicine and Pharmacy of Cluj-Napoca, 400012 Cluj-Napoca, Romania; drnicolaecrisan@gmail.com; 20Urology Unit, Department of Medical, Oral and Biotechnological Sciences, “SS. Annunziata” Hospital, 66100 Chieti, Italy; mic.marchioni@gmail.com (M.M.); luigischips@hotmail.com (L.S.); 21Division of Urology, Department of Surgical Sciences, University of Studies of Torino, 10126 Turin, Italy; francesco.soria@unito.it; 22Department of Urology, Campus Bio-Medico University, 00128 Rome, Italy; r.papalia@unicampus.it; 23Urology Unit, Department of Neurosciences, Sciences of Reproduction and Odontostomatology, University of Naples “Federico II”, 80125 Naples, Italy; felice.crocetto@unina.it (F.C.); biagio.barone@unina.it (B.B.); 24Department of Urology, University of Catania, 95123 Catania, Italy; giorgioivan.russo@unict.it; 25Department of Urology, F. Miulli Regional Hospital, Acquaviva delle Fonti, 70021 Bari, Italy; g.ludovico@miulli.it; 26Department of Urology, Medical University of Vienna, 1090 Vienna, Austria; 27Department of Urology, University of Naples “Federico II”, 80125 Naples, Italy; vincenzo.mirone@ieo.it

**Keywords:** non muscle invasive bladder cancer, Bacillus Calmette–Guérin, modified Glasgow prognostic score

## Abstract

Background: A systemic inflammatory marker, the modified Glasgow prognostic score (mGPS), could predict outcomes in non-muscle-invasive bladder cancer (NIMBC). We aimed to investigate the predictive power of mGPS in oncological outcomes in HG/G3 T1 NMIBC patients undergoing Bacillus Calmette–Guérin (BCG) therapy. Methods: We retrospectively reviewed patient’s medical data from multicenter institutions. A total of 1382 patients with HG/G3 T1 NMIBC have been administered adjuvant intravesical BCG therapy, every week for 3 weeks given at 3, 6, 12, 18, 24, 30 and 36 months. The analysis of mGPS for recurrence and progression was performed using multivariable and univariable Cox regression models. Results: During follow-up, 659 patients (47.68%) suffered recurrence, 441 (31.91%) suffered progression, 156 (11.28%) died of all causes, and 67 (4.84%) died of bladder cancer. At multivariable analysis, neutrophil to lymphocyte ratio [hazard ratio (HR): 7.471; *p* = 0.0001] and erythrocyte sedimentation rate (ESR) (HR: 0.706; *p* = 0.006 were significantly associated with recurrence. mGPS has no statistical significance for progression (*p* = 0.076). Kaplan–Meier survival analysis showed a significant difference in survival among patients from different mGPS subgroups. Five-year OS was 93% (CI 95% 92–94), in patients with mGPS 0, 82.2% (CI 95% 78.9–85.5) in patients with mGPS 1 and 78.1% (CI 95% 60.4–70) in mGPS 2 patients. Five-year CSS was 98% (CI 95% 97–99) in patients with mGPS 0, 90% (CI 95% 87–94) in patients with mGPS 1, and 100% in mGPS 2 patients. Limitations are applicable to a retrospective study. Conclusions: mGPS may have the potential to predict recurrence in HG/G3 T1 NMIBC patients, but more prospective, with large cohorts, studies are needed to study the influence of systemic inflammatory markers in prediction of outcomes in NMIBC for a definitive conclusion.

## 1. Introduction

Bladder cancer is the 10th most common diagnosed cancer in the world, with estimated 573,000 new cases and 213,000 deaths in 2021. It is more common in men with incidence and mortality rates of 9.5 and 3.3/100.000 in men, four times higher than in women [1]. The high prevalence of tobacco smoking, infections and occupational exposures may be major causes for bladder cancer [2,3]. Non-muscle invasive bladder cancer (NMIBC) patients treated with 1–3 years of maintenance Bacillus Calmette–Guérin (BCG), especially high-grade (HG)/G3 T1 patients are among the highest risk for disease-progression, with 1- and 5-year rates of 11.4% and 19.8%, respectively [4,5,6]. In patients with HG/G3 T1 tumors after adjuvant instillations of BCG, the risk of recurrence can be assessed using clinical and pathologic variables such as tumor size, age, smoking status and presence of carcinoma in situ [7], gender [8], multifocality of tumors [9], neutrophil-to-lymphocyte ratio (NLR) [10] and the diagnosis of HG/G3 T1 tumors on re-transurethral resection of bladder tumor (TURBT) [11]. Body mass index (BMI) [9,12], as well as lymph-vascular invasion (LVI) and LVI at re-TURBT [13,14], affects the risk of recurrence and progression. It is well known that systemic inflammation plays a key role in solid tumors and measuring C-reactive protein (CRP) represents a rapid and relatively cheap prognostic circulating biomarker [15]. CRP has been extensively evaluated in urothelial carcinoma in patients subject to TURBT [16], chemotherapy [17] or radical cystectomy [18]. CRP has also been studied and incorporated along with albumin, into a risk score model, the modified Glasgow prognostic score (mGPS), which showed an independent prognostic value in patients undergoing radical cystectomy [19] and radiotherapy [20]. The clinical role and the possibility of using such a prognostic score has been studied in muscle invasive bladder cancer [19,20,21,22,23,24] and NMIBC [25,26,27]. Therefore, we aimed to investigate its prognostic value in HG/G3 T1 NMIBC patients subjected to TURBT and re-TURBT followed by three-year intravesical BCG instillations.

## 2. Materials and Methods

A total of 1382 NMIBC (HG/G3 T1) patients, from 1 January 2002 till 31 December 2012, that underwent TURBT from 13 healthcare centers and research institutions, were enlisted in a retrospective longitudinal study. Inclusion criteria were as follows: Patients with intravesical BCG therapy continued with maintenance instillations. Clinical, pathological and demographical data were uploaded into an electronic database. Patients with incomplete BCG therapy (induction without maintenance) were not enrolled in the study. Histology assessment has been performed at each hospital. Classification of tumors has been performed according to the TNM system of Union for International Cancer Control (UICC) and to the 1973 World Health Organization (WHO) grading system. The strategy for re-TURBT was comprised in resection of the previous scar and base, bladder neck for CIS and any suspicious patches. All patients had re-TURBT performed within 4–6 weeks and then intravesical BCG therapy was administered as an adjuvant therapy. A six-week course of intravesical BCG induction therapy was recommended, followed by intravesical BCG, every week for three weeks, and then up to three years after the start of the instillations. The whole urothelial tract was evaluated for concomitant carcinoma [28]. Serum data, including inflammatory markers, were collected one month before surgery. All patients signed a written informed consent. The mGPS was calculated as previously described [29]. The score was assigned as 0 for patients with CRP <10 mg/L and no abnormalities of the serum albumin levels. Further, score 1 has been given to patients with elevated CRP (>10 mg/L), and elevated CRP (>10 mg/L) plus hypoalbuminaemia (<3.5 g/dL), as score 2. The study endpoints were the time to recurrent disease, the time to progressive disease, the overall survival (OS) and the cancer specific survival (CSS).

### 2.1. Follow-Up

The follow-up included cystoscopy and urinary cytology every 3 months in compliance with the EAU guidelines [28]. The appearance of any tumor meant recurrence and muscle-invasive disease was considered progression. Patients with muscle-invasive bladder cancer on re-TURBT and those who recurred or progressed under BCG therapy were subjected to radical cystectomy [30]. Patients with muscle-invasive bladder cancer after re-TURBT were not further included in the analysis. 

### 2.2. Statistical Methods 

All continuous variables were checked for normality using the Kolmogorov–Smirnoff test. Differences between categories were tested using ANOVA for variables with a normal distribution and homogeneity of variances, the Kruskal–Wallis for variables not normally distributed and Chi-squared test for categorical variables. The Bonferroni/Dunns correction was used for post hoc comparisons between pairs of categories. Outcomes of interest were recurrence, progression, CSS, and OS. Kaplan–Meier curves were applied to calculate the association between mGPS and recurrence-free survival (RFS), progression-free survival (PFS), CSS and OS. Log-rank test was used to verify statistical significance between curves. Multivariable Cox regression analysis was performed to identify predictive factors of recurrence and progression, using the variables collected. All statistical analyses were carried out using SPSS for Windows version 22.0 (SPSS, Inc., Chicago, IL, USA). For all comparisons, the significance level was set to *p* < 0.05 for differences between groups.

## 3. Results

### 3.1. Clinical and Pathological Features for the Whole Cohort of Patients 

The median age was 69.87 years [interquartile range (IQR) 60.16–79.58]. Regarding the mGPS distribution, 1001 patients (72.4%) had an mGPS equal to 0, 341 patients (24.7%) had an mGPS of 1 and 40 patients (2.9%) had an mGPS of 2. Gender distribution was 80.7% (1115) males and 19.3% (267) females. From the whole cohort, 35 patients (2.53%) received an early single instillation immediately after TURBT, according to EAU guidelines. In particular, 3 (0.21%) received Gemcitabin, 25 (1.81%) Epirubicin and 7 (0.51%) Mytomicin. After re-TURBT, 341 (24.67%) patients showed residual high grade NMIBC, and 1041 patients (75.32%) were negative. All patients received BCG immunotherapy. The median CRP levels were 2 mg/dL (IQR 0.01–125.0) and median albumin levels of 4.17 g/dL (IQR 3.65–4.69). Clinical and pathological data are presented in Table 1.

### 3.2. Association of Clinical and Pathologic Features with mGPS in 1382 Patients Treated with BCG after Primary HG/G3 T1 NMIBC Included in Follow-Up

The analysis of the association of clinical and pathologic features with mGPS in 1382 patients showed that mGPS is significantly associated with smoking status (*p* = 0.001), the multifocality of tumors (*p* = 0.014) and tumor size (*p* = 0.011) (Table 2).

### 3.3. Association of mGPS with Recurrence and Progression

Within a median time of 27 months (IQR 8–35), 659 (47.68%) patients relapsed, and within a median time of 22 months (IQR 36–58) 441 (31.91%) patients underwent progression. The mean time for recurrence in the mGPS 0 was 61.7 months (CI 95% 58.2–65.1), higher than the mean time for recurrence for mGPS 1 [53.1 months (CI 95% 46.5–59.7)] and lower than the mean time to recurrence for mGPS 2, [68.2 months (CI 95% 50.1–86.3), *p* < 0.0001]. In the multivariable model, we found that NLR (Neutrophil to Lymphocyte Ratio) [hazard ratio (HR): 7.471; *p* = 0.0001], ESR (HR: 0.762; *p* = 0.006) were significantly associated with recurrence. mGPS 1 (HR: 1.417; *p* = 0.0001) is significantly associated with an increased risk of recurrence (Table 3). Kaplan–Meier survival analysis showed that patients with mGPS 1 had significantly reduced recurrence free survival (RFS) [32.4% (CI 95% 28.6–36.2)], as compared to mGPS 0 [42.9% (CI 95% 40.9–44.9), *p* < 0.001] (Figure 1A). 

Kaplan–Meier survival analysis showed that mGPS 0 patients and mGPS 1 and 2 patients did not have a significantly shorter progression-free survival (PFS), *p* = 0.76. The mean time for progression for mGPS 0 was 82.6 months (CI 95% 58.2–65.1), higher than the mean progression for mGPS 1 of 79.1 months (CI 95% 46.5–59.7) and lower for mGPS 2 of 86 months (CI 95% 50.1–86.3), with the differences not reaching statistical significance (*p* = 0.76). The five-year progression rate for mGPS 0 was 63.3% (CI 95% 61.3–65.3), for mGPS 1 of 57.1% (CI 95% 53.1–61.1) and for mGPS 2 of 69.2% (CI 95% 60.4–78) (Figure 1B).

At multivariable analysis ESR (HR: 0.692; *p* < 0.003) was significantly associated with progression. mGPS 1 was significantly associated with progression in the univariable (HR: 1.324; *p* = 0.015) and in the multivariable analysis (HR: 1.361; *p* = 0.011) and mGPS 2 was not significantly associated with an increased risk of progression, both in the univariable (*p* = 0.648) and in the multivariable analysis (*p* = 0.731). Table 4 comprises the Cox regression analyses for the study population.

### 3.4. Association of the mGPS with Overall and Cancer-Specific Survival

Within a median follow-up of 44 months (IQR: 36–58), 156 (11.28%) died due to general causes, while 67 (4.84%) died of BC. Kaplan–Meier survival analysis showed a significant difference in survival among patients from different mGPS subgroups. Mean overall survival for patients having mGPS of 0 was 107.3 months (CI 95% 105.1–109.5) and it was higher than the mean survival for patients with mGPS 1 of 102.1 months (CI 95% 96.2–108), and for mGPS 2 of 101.5 months (CI 95% 89.7–113.4), *p* = 0.015. Five-year OS was 93% (CI 95% 92–94), in patients with mGPS 0, 82.2% (CI 95% 78.9–85.5) in patients with mGPS 1 and 78.1% (CI 95% 60.4–70) in mGPS 2 patients. Five-year CSS was 98% (CI 95% 97–99) in patients with mGPS 0, 90% (CI 95% 87–94) in patients with mGPS 1, and 100% in mGPS 2 patients (Figure 2A,B).

## 4. Discussion

In this retrospective longitudinal study, we evaluated the association between mGPS and the clinical outcome of NMIBC patients undergoing TURBT plus adjuvant BCG intravesical instillation therapy. We found that mGPS was a significant predictor of clinical outcome in patients with HG/G3 T1 NMIBC. Some researchers previously described the association of inflammatory markers, mGPS and the outcomes of NMIBC patients [26,27]. The EORTC risk tables [7] and the EAU risk categories [31], are currently used for risk stratification and prognosis prediction in NMIBC patients. For BCG treated patients, the CUETO model [6] is better in predicting oncological outcomes compared to EORTC risk tables, but there is no standardization for risk assessment and treatment in such patients, thus the precision in stratifying these patients is a clinical need. Ferro et al. [32] demonstrated that subjects with an mGPS 2 had a significantly shorter median RFS compared to subjects with mGPS 1 or with mGPS 0 (*p* < 0.001) and the association between mGPS and RFS was confirmed by weighted multivariable Cox model. Kimura et al. [26], found that mGPS 1 and 2 were both independently associated with worse PFS, compared to mGPS 0. In our study mGPS 2 was not found to be significantly associated with RFS or PFS in the univariable (*p* = 0.531, *p* = 0.648, respectively) and multivariable (*p* = 0.321, *p* = 0.731, respectively) analysis. Different factors could explain these discrepancies with our results. Firstly, the low number of patients (2.9%) compared to the overall cohort could lead to a decrease in the statistical analysis strength and possibly lowering definitive clinical conclusions. Secondly, we performed reTURBT in all patients involved, possibly impacting the prognosis and the findings related to mGPS. Lastly, in the study of Kimura et al., only 279 patients on 503 EAU high-risk patients were treated with BCG, while we reported an overall higher number of EAU high-risk patients. NLR is well known as an independent predictor of disease recurrence and progression in NMIBC treated with BCG patients [10]. NLR before treatment was correlated with both oncological outcomes and survival outcome in NMIBC patients undergoing initial intravesical BCG treatment after TURBT [33]. Our results enforce the fact that NLR is a suitable pretreatment marker for prognosis in NMIBC patients treated with BCG. In our cohort of 1382 patients, we aimed to evaluate systemic inflammation by using mGPS and we found that mGPS was associated with smoking status (*p* < 0.001), multifocality (*p* = 0.014) and tumor size (*p* = 0.011). Among the clinical and pathological features, no significant association was observed between mGPS and age, gender, concomitant CIS, recurrence, progression, OS and CSS. Based on these results, we showed that the effects of tobacco smoke have a detrimental effect on both inflammation and oxidative stress which may lead to activation of angiogenesis and tumor progression, according to literature data [27,34,35]. Moreover, our data from Cox regression analysis showed that patients with mGPS of 1 were more likely to experience disease recurrence (univariable HR: 1.403; *p* < 0.0001; (multivariable HR: 1.542; *p* < 0.0001), compared to mGPS 0 patients. We identified an association of the mGPS 1 with progression both in the univariable and multivariable analysis in NMIBC patients treated with TURBT and intravesical BCG therapy (RFS-HR: 1.417; *p* = 0.0001 and PFS-HR: 1.361; *p* = 0.011). Finally, although EAU recommends immediate instillation after TURBT for all superficial bladder tumors, the efficacy of a single immediate Mytomicin instillation after TURBT is still controversial. However, this treatment does not impact the indication to reTURBT nor the efficacy and the response of BCG therapy [36]. Our study has several limitations. First, the study’s retrospective design and separate pathology tissue examinations could lead to patients’ selection bias. Second histopathological examination was not reviewed in a centralized way, and mGPS was not calculated after TURBT. Third, the non-standardization of TURBT and the differences in BCG intravesical instillations scheduling and adjuvant therapy performed can lead to low assessment of maintenance completion of intravesical therapy. Collectively, our findings suggested that the mGPS was a prognostic factor in muscle invasive bladder cancer followed by cystectomy, but our cohort is comprised only with NMIBC patients’ mGPS can be used to early predict recurrence being highly cost-effective. Being a retrospective study external validation is needed and future well designed research is mandatory to validate the prediction validity of mGPS.

## 5. Conclusions

The mGPS has the potential to be associated with the risk of recurrence in HG/G3 T1 bladder cancer patients treated with standard combination of TURBT and BCG instillations. Inflammatory and nutritional status can influence the oncological outcomes of such patients and possible the efficacy of BCG as immunotherapy. Therefore, further research is needed to warrant the prognostic value of mGPS for better risk stratification in this category of bladder cancer patients.

## Figures and Tables

**Figure 1 diagnostics-12-00586-f001:**
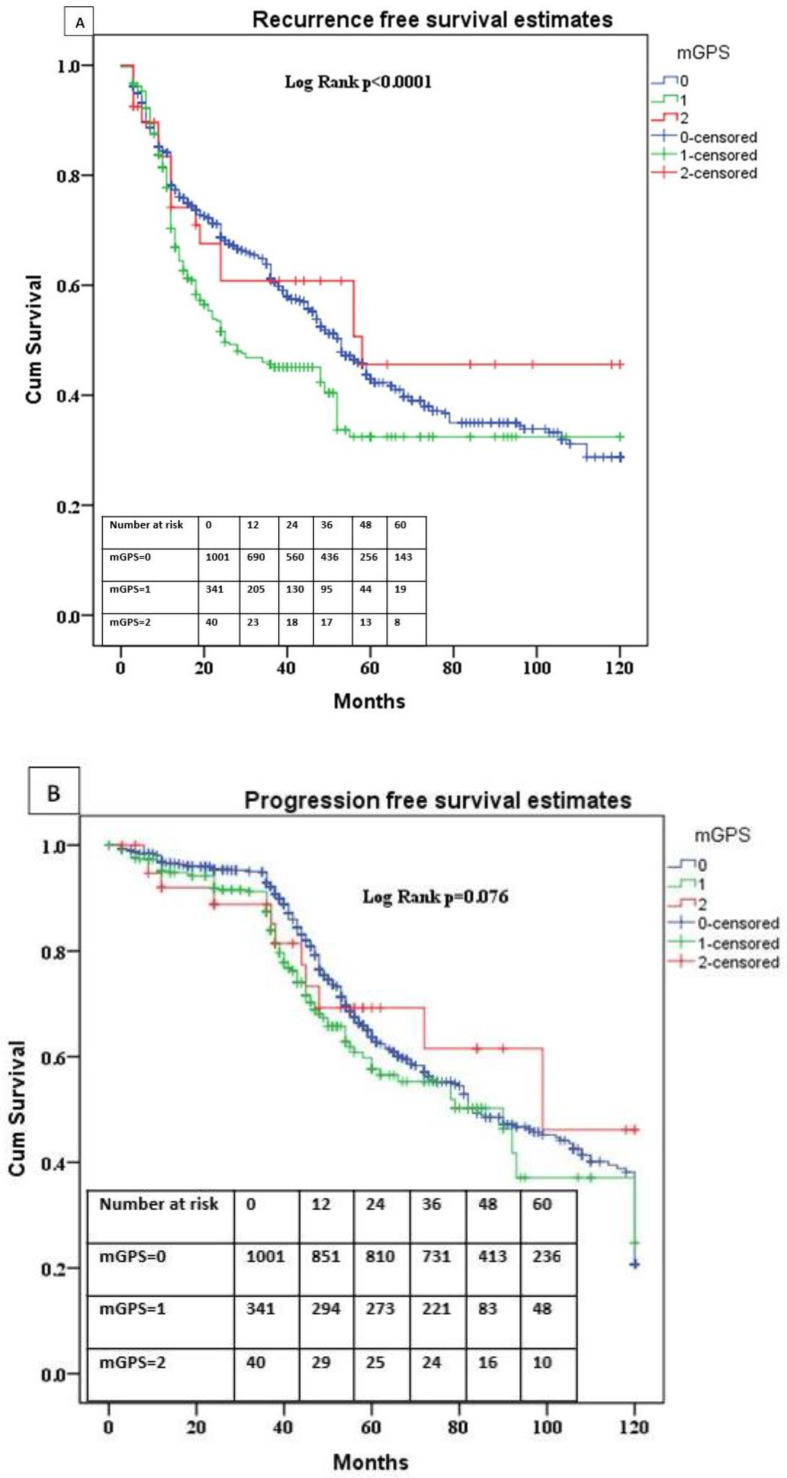
(**A**) Comparison of recurrence-free survival according to mGPS status. (**B**) Comparison of progression-free survival according to mGPS status.

**Figure 2 diagnostics-12-00586-f002:**
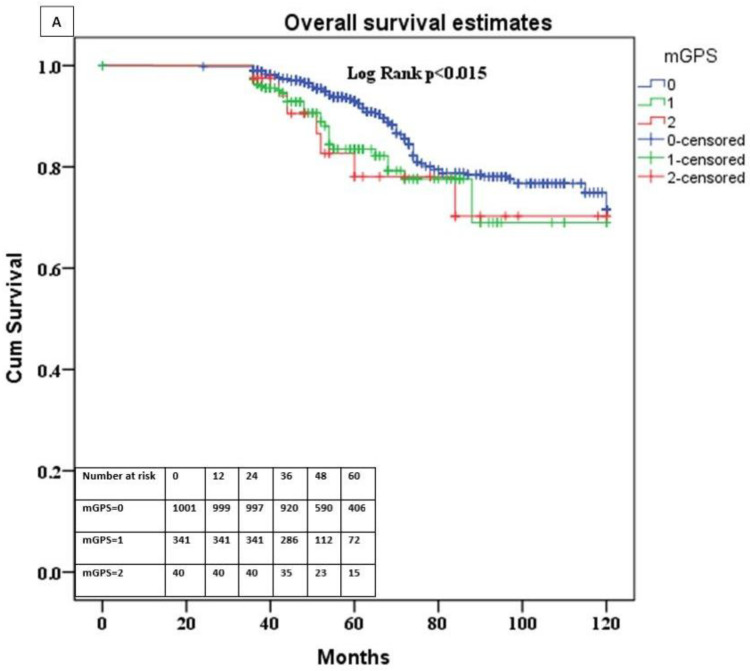
(**A**) Comparison of overall survival estimates according to mGPS status. (**B**) Comparison of cancer-specific survival estimates according to mGPS status.

**Table 1 diagnostics-12-00586-t001:** Clinical characteristics of the patient population.

Overall Cohort (*n* = 1382)	
Age (years)	69.87 ± 9.71
Gender (males/females)	1115/267 = 80.7%/19.3%
BMI (kg/m^2^)	26.8 ± 3.79
Smoking status	
Yes	784 (56.7%)
No	598 (43.3%)
Number of cigarettes/day	10 (0–65)
Number of years of smoking	20 (0–70)
Multifocality-yes	605 (43.8%)
Tumor size > 3 cm	887 (64.2%)
Concomitant CIS	224 (16.2%)
LVI	212 (15.3%)
LVI at re-TURB	85 (6.2%)
ESR (mm/h)	12 (1–109)
CRP (mg/dL)	2 (0.01–125.0)
Albumin (g/dL)	4.17 ± 0.52
mGPS	
0	1001 (72.4%)
1	341 (24.7%)
2	40 (2.9%)

BMI—body mass index, CIS-carcinoma in situ, CRP—C-reactive protein, LVI—lymph-vascular invasion, mGPS—modified Glasgow prognostic score, ESR—erythrocyte sedimentation rate.

**Table 2 diagnostics-12-00586-t002:** Association of clinical and pathologic features with mGPS in 1382 patients treated with BCG after HG/G3 T1 NMIBC.

	mGPS 0	mGPS 1	mGPS 2	*p* Value
Total, *n* (%)	1001 (72.4%)	341 (24.7%)	40 (2.9%)	-
Age (years) Mean (SD)	69.97 ± 9.68	69.74 ± 9.62	68.65 ± 11.24	0.56 *
Gender, *n* (%)				
Male	810 (72.6%)	277 (24.8%)	28 (2.5%)	0.20 **
Female	191 (71.8%)	63 (23.7%)	12 (4.5%)	
Smoking status				
Yes	285 (73.1%)	89 (22.8%)	16 (4.1%)	0.001 **
No	404 (67.6%)	180 (30.1%)	14 (2.3%)	
Multifocality, *n* (%)				
Unifocal	587 (75.5%)	170 (21.9%)	20 (2.6%)	0.014 **
Multifocal	414 (68.4%)	171 (28.3%)	20 (3.3%)	
Size, *n* (%)				
<3 cm	381 (77.0%)	99 (20.0%)	15 (3.0%)	0.011 **
≥3 cm	620 (69.9%)	242 (27.3%)	25 (2.8%)	
Concomitant CIS, *n* (%)				
Yes	165 (73.7%)	54 (24.1%)	5 (2.2%)	0.76 **
No	836 (72.2%)	287 (24.8%)	35 (3.0%)	
Survival outcomes				
Recurrence, *n* (%)				
Yes	470 (71.3%)	173 (26.3%)	16 (2.4%)	0.30 **
No	531 (73.4%)	168 (23.2%)	24 (3.3%)	
Progression, *n* (%)				
Yes	332 (75.3%)	98 (22.2%)	11 (2.5%)	0.27 **
No	669 (71.1%)	243 (25.8%)	29 (3.1%)	
Death	113 (72.4%)	36 (23.1%)	7 (4.5%)	0.43 **
Death due to BC	47 (70.1%)	20 (29.9%)	0 (0.0%)	0.22 **

* Anova Test, ** Chi-squared Test. BCG—Bacillus Calmette–Guérin, NMIBC—non-muscle-invasive bladder cancer, CIS—carcinoma in situ, BC—bladder cancer, mGPS—modified Glasgow prognostic score.

**Table 3 diagnostics-12-00586-t003:** Univariable and multivariable Cox regression analyses predicting recurrence in 1382 patients with HG/G3 T1 NMIBC treated with BCG.

Variables	Recurrence-Free Survival					
	Univariable			Multivariable		
	HR	95% CI	*p* Value	HR	95% CI	*p* Value
Age cont.	0.840	0.716–0.986	0.033	1.167	0.965–1.411	0.112
Gender (male vs. female)	1.074	0.980–1.177	0.125	0.82	0.656–1.022	0.078
Size (<3 vs. ≥3) cm	1.154	1.082–1.355	0.042	0.984	0.815–1.189	0.870
Multifocality (single vs. multiple)	1.177	1.009–1.373	0.038	1.146	0.953–1.379	0.147
Concomitant CIS (no vs. yes)	1.138	0.932–1.389	0.201	0.906	0.715–1.149	0.416
NLR	6.864	5.234–9.002	0.001	7.471	5.394–10.346	0.0001
ESR (mm/h)	0.911	0.768–1.080	0.282	0.706	0.628–0.924	0.006
mGPS						
0	ref					
1	1.403	1.177–1.673	0.0001	1.417	1.187–1.692	0.0001
2	0.853	0.518–1.404	0.531	1.239	0.813–2.328	0.321

BCG—Bacillus Calmette–Guérin, NMIBC—non-muscle-invasive bladder cancer, CIS—carcinoma in situ, mGPS—modified Glasgow prognostic score, ESR—erythrocyte sedimentation rate, NLR—neutrophil–lymphocyte ratio.

**Table 4 diagnostics-12-00586-t004:** Cox regression analyses predicting progression in 1382 patients with primary HG/G3 T1 NMIBC treated with BCG.

Variables	Progression-Free Survival					
	Univariable			Multivariable		
	HR	95% CI	*p* Value	HR	95% CI	*p* Value
Age cont.	0.749	0.585–0.959	0.022	1.085	0.859–1.370	0.493
Gender (male vs. female)	0.960	0.849–1.085	0.509	0.864	0.649–1.149	0.314
Size (<3 vs. ≥3) cm	1.136	0.934–1.382	0.201	1.078	0.862–1.347	0.512
Multifocality (single vs. multiple)	1.195	0.987–1.447	0.067	1.217	0.976–1.518	0.081
Concomitant CIS (no vs. yes)	1.243	1.068–1.595	0.044	1.065	0.798–1.422	0.667
NLR	1.010	0.827–1.240	0.904	1.014	0.800–1.285	0.909
ESR (mm/h)	1.173	0.969–1.421	0.102	0.692	0.532–0.874	0.003
mGPS						
0	ref					
1	1.324	1.044–1.663	0.015	1.361	1.072–1.727	0.011
2	0.869	0.477–1.586	0.648	0.899	0.492–1.645	0.731

BCG—Bacillus Calmette–Guérin, NMIBC—non-muscle-invasive bladder cancer, CIS—carcinoma in situ, BC—bladder cancer, mGPS—modified Glasgow prognostic score, ESR—erythrocyte sedimentation rate, NLR—neutrophil–lymphocyte ratio.

## Data Availability

The data presented in this study are available on request from the corresponding author. The data are not publicly available due to privacy regulations.

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
