# Peer review of "Modified Glasgow Prognostic Score as a Predictor of Recurrence in Patients with High Grade Non-Muscle Invasive Bladder Cancer Undergoing Intravesical Bacillus Calmette–Guerin Immunotherapy"

_diagnostics, 2022, doi:10.3390/diagnostics12030586_

Round 1
Reviewer 1 Report
It is an interesting study but some points should be clarified .
- It the result section: "From the whole cohort, 35 patients (2.53%) had received other intravesical chemotherapy instillation, 3 (0.21%) Gemcitabin, 10 (0.72%) Epirubicin, 15 (1.08%) Farmacorubicin and 138 (9.98%) Mitomycin. " The number seems incorrect ( 3+10+15+138 = 168. 168 use other IVI C/T ) , please explain ?
- Please standardize the figures size .
- IN discussion : "Ferro et al. demonstrated that subjects with an mGPS 2 had a significantly shorter median RFS compared to subjects with mGPS 1 or with mGPS 0 (p<0.001) and the association between mGPS and RFS was confirmed by weighted multivariable Cox model. Kimura et al. , found that mGPS 1 and 2 were both independently associated with worse PFS," but your data fail to support mGPS 2 associated with poor recurrence and progression than mGPS 0 and 1. You explain the cause may be smaller number of mGPS 2. However, both previous two studies also showed smallest group number in mGPS2. Please explain and interpret !
Author Response
REVIEWER 1
- It the result section: "From the whole cohort, 35 patients (2.53%) had received other intravesical chemotherapy instillation, 3 (0.21%) Gemcitabin, 10 (0.72%) Epirubicin, 15 (1.08%) Farmacorubicin and 138 (9.98%) Mitomycin. The number seems incorrect (3+10+15+138 = 168. 168 use other IVI C/T), please explain?
We thank the reviewer for her/his suggestion. We checked and corrected the total number of patients who underwent other intravesical therapies. In addition, we clarified the patients who received Mytomicin as early single instillation after TURBT (and then proceed to the classic BGC induction/maintenance scheme). We further added in the discussion the rationale and the absence of impact of this practice on efficacy and response to BCG therapy.
- Please standardize the figures size.
We thank the reviewer for her/his suggestion. We modified figures accordingly
- In discussion : "Ferro et al. demonstrated that subjects with an mGPS 2 had a significantly shorter median RFS compared to subjects with mGPS 1 or with mGPS 0 (p<0.001) and the association between mGPS and RFS was confirmed by weighted multivariable Cox model. Kimura et al., found that mGPS 1 and 2 were both independently associated with worse PFS," but your data fail to support mGPS 2 associated with poor recurrence and progression than mGPS 0 and 1. You explain the cause may be smaller number of mGPS 2. However, both previous two studies also showed smallest group number in mGPS2. Please explain and interpret!
We thank the reviewer for her/his suggestion. We reported the following in order to explain the underlined discrepancies:
“Different factors could explain these discrepancies with our results. Firstly, the low number of patients (2.9%) compared to the overall cohort could lead to a decrease in the statistical analysis strength and possibly lowering definitive clinical conclusions. Secondly, we performed reTURBT in all patients involved, possibly impacting the prognosis and the findings related to mGPS. Lastly, in the study of Kimura et al., only 279 patients on 503 EAU high-risk patients were treated with BCG, while we reported an overall higher number of EAU high-risk patients.”
Reviewer 2 Report
For patients who have received other intravesical chemotherapies(Gemcitabine, Epirubicin, Pharmacorubicin and Mitomycin.
Are they excluded from the analysis? If not, why not?
Because this type of intravesical therapy is completely
different from BCG.
Author Response
REVIEWER 2
- For patients who have received other intravesical chemotherapies (Gemcitabine, Epirubicin, Pharmacorubicin and Mitomycin. Are they excluded from the analysis? If not, why not? Because this type of intravesical therapy is completely different from BCG
We thank the reviewer for her/his suggestion. We checked and corrected the total number of patients who underwent other intravesical therapies. Regarding other chemotherapies, the sample size was very limited and therefore analysis was not feasible. Regarding Mytomicin, as reported in the discussion:
“Finally, although EAU recommends immediate instillation after TURBT for all superficial bladder tumors, the efficacy of a single immediate Mytomicin instillation after TURBT is still controversial. However, this treatment does not impact the indication to reTURBT nor the efficacy and the response of BCG therapy”
Round 2
Reviewer 1 Report
improved after revision